# HAT: Hypergraph analysis toolbox

**Joshua Pickard** [1,2], **Can Chen** [3], **Rahmy Salman** [4], **Cooper Stansbury** [1], **Sion Kim** [4], **Amit Surana** [5], **Anthony Bloch** [6], **Indika Rajapakse** [1,2,6] *

**1** Department of Computational Medicine and Bioinformatics, University of Michigan, Ann Arbor, Michigan, United States of America, **2** iReprogram, Inc., Ann Arbor, Michigan, United States of America, **3** Channing Division of Network Medicine, Department of Medicine, Brigham and Women's Hospital and Harvard Medical School, Boston, Massachusetts, United States of America, **4** Department of Electrical Engineering and Computer Science, University of Michigan, Ann Arbor, Michigan, United States of America, **5** Raytheon Technologies Research Center, East Hartford, Connecticut, United States of America, **6** Department of Mathematics, University of Michigan, Ann Arbor, Michigan, United States of America

* indikar@umich.edu

**Data Availability Statement:** All relevant code and documentation may be found at the software main page https://hypergraph-analysis-toolbox.readthedocs.io/ Additional open source distributions of HAT are available on GitHub

## Abstract

Recent advances in biological technologies, such as multi-way chromosome conformation capture (3C), require development of methods for analysis of multi-way interactions. Hypergraphs are mathematically tractable objects that can be utilized to precisely represent and analyze multi-way interactions. Here we present the Hypergraph Analysis Toolbox (HAT), a software package for visualization and analysis of multi-way interactions in complex systems.

## Author summary

Classical networks typically focus on pairwise interactions and may overlook the intricate higher-order, multi-way interactions that occur among groups of nodes within a network. Our research has delved into the structural and dynamic characteristics of hypergraphs, which can effectively capture multi-way network interactions across various domains and data types. In this article, we introduce the Hypergraph Analysis Toolbox (HAT), a software package encompassing a range of techniques to identify, investigate, and visualize multi-way interactions in biological data.

This is a *PLOS Computational Biology* Software paper.

## Introduction

Network science is a powerful framework for studying complex systems. However, recent work highlights the limitations of classical methods in networks, which only consider pairwise interactions between nodes to describe group interactions. Use of hypergraphs, in which an edge can connect more than two nodes, has therefore emerged as a new frontier in network science [1–3].

(https://github.com/Jpickard1/Hypergraph-Analysis-Toolbox), PyPI (https://pypi.org/project/HypergraphAnalysisToolbox/), and MATLAB Central (https://www.mathworks.com/matlabcentral/fileexchange/121013-hat-hypergraph-analysis-toolbox), which are all linked on the main page. All issue and bug tracking for the software are handled through GitHub issues (https://github.com/Jpickard1/Hypergraph-Analysis-Toolbox/issues) which is also referenced from the main page. Tutorials on usage of the HAT are available through Google CoLab and MATLAB Online (https://hypergraph-analysis-toolbox.readthedocs.io/en/latest/tutorials.html).

**Funding:** This work is supported in part by the Air Force Office of Scientific Research (AFOSR) awards FA9550-18-1-0028 and FA9550-22-1-0215 (IR), NSF DMS2103026 (AB), and a MathWorks Fellowship to the Rajapakse Lab (IR). The funders had no role in study design, data collection and analysis, decision to publish, or preparation of the manuscript.

**Competing interests:** The authors have declared that no competing interests exist.

Chromosome conformation capture (3C) methods identify physical interactions ("contacts") between genomic loci [4, 5]. While classical 3C is pairwise, recent advancements capture multi-way chromatin interactions via proximity ligation (see Pore-C in S1 File) [6], Split-Pool Recognition of Interactions by Tag Extension (SPRITE) [7, 8], or multi-contact 3C (MC-3C) [9]. However, the investigation and biological interpretation of these multi-way contacts is hampered by scarcity of methods for multi-way data [6, 10]. Hypergraphs are a mathematically tractable extension of graph theory that precisely represent multi-way interactions (See Hypergraphs in S1 File) [2]. We introduce the Hypergraph Analysis Toolbox (HAT), a general purpose software for the analysis of multi-way interactions and higher-order structures. HAT contains both well-studied and novel mathematical methods for hypergraph analysis in both MATLAB and Python.

Motivated to investigate Pore-C data, HAT is designed as a versatile software for hypergraph analysis. While there are several robust libraries for graph analysis, most hypergraph software is not multi-faceted and targets specific problems, such as hypergraph partitioning or clustering (Table 1). As a general purpose tool, the algorithms implemented in HAT address hypergraph construction, visualization, and the analysis of structural and dynamic properties. HAT is the first software to utilize tensor algebra for hypergraph analysis [11–13], and it contains recently developed methods for hypergraph similarity measures [13]. HAT is open source, standardized across MATLAB (version 2021b onward) and Python (version 3.7 onward) implementations, and is documented at https://hypergraph-analysis-toolbox.readthedocs.io, where it will continue to be maintained and developed.

For ease of use, the MATLAB and Python implementations are functionally independent but syntactically similar. The software may be installed from the online documentation, GitHub, or via PIP and the MathWorks file exchange for the respective Python and MATLAB implementations.

## Materials and methods

HAT can visualize and analyze multi-way interactions. The incidence matrix is the primary representation of hypergraphs in HAT (Fig 1b) [10, 23]. HAT targets the following hypergraph features and problems: (1) construction from data [24–26], (2) expansion and numeric representation [27–29], (3) characteristic structural properties (such as entropy [11], centrality [30], distance [13], and clustering coefficients [11]), (4) controllability [12], and (5) similarity measures [13]. The workflow for using HAT is outlined in Fig 1e.

### Construction from data

There are two approaches for constructing a hypergraph from data (see Hypergraphs in S1 File). Data formats with explicit multi-way interactions, such as Pore-C are directly input to HAT for hypergraph construction. However, the vast majority of data are either pairwise

**Table 1. Comparison of HAT to well-documented hypergraph libraries.** There are several other notable hypergraph software not listed in the table [19–22].

| Software | Language | Features |
| --- | --- | --- |
| Hypergraph Analysis Toolbox | MATLAB and Python | construction, visualization, expansion, similarity measures, centrality, entropy, tensor- based analysis, controllability |
| HyperNetX [14] | Python | clustering, visualization, homology, clustering |
| HALP [15] | Python | directed hypergraphs, walks, partitioning |
| hMETIS [16] | C/C++ | partitioning, implemented in parallel |
| Phoenix [17] | C/C++ | clustering, implemented in parallel |
| HyperG [18] | R | hypergraph creation, clustering, graph based representations and calculations, visualization |

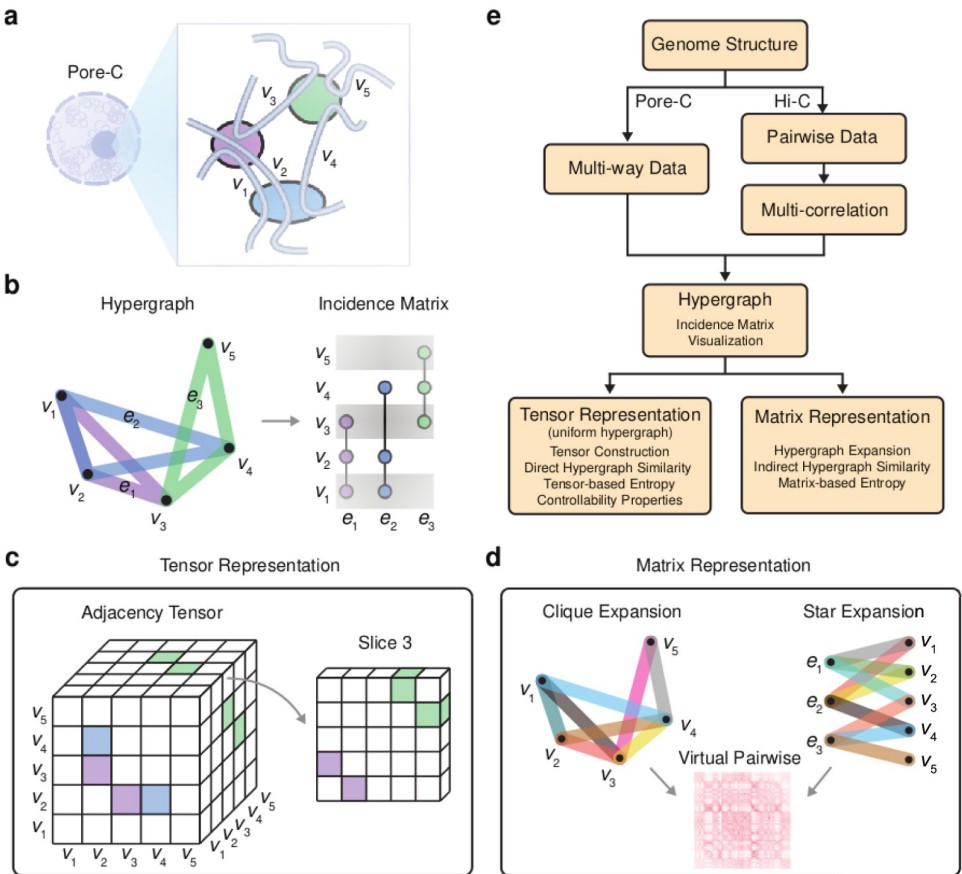

**Fig 1. HAT overview. a.** The Pore-C assay identifies multi-way chromatin strand colocalization within the nucleus [6]. **b.** Hypergraph representation of Pore-C is drawn where each chromatin strand is represented as a vertex and the multi-way contacts are hyperedges. This is depicted as both a hypergraph and an incidence matrix. **c.** For multi-way contacts of uniform size, hypergraphs are numerically represented as an adjacency tensor or multi-dimensional matrix. **d.** Multi-way structure are decomposed with clique and star expansions that generate virtual pairwise contacts [6]. **e.** The workflow of HAT to construct hypergraphs from data, visualize, represent numerically, and computations available for each representation are outlined as a flowchart.

observations (e.g., Hi-C) or do not contain either pairwise or multi-way interactions (e.g., sequencing data), so we implemented three measures to infer multi-way relationships based on multi-correlation measures [24–26]. HAT constructs hyperedges by setting a minimum threshold for the multi-correlation to be considered a hyperedge.

## Expansion and numerical representation

For uniform hypergraphs, the adjacency, degree, and Laplacian tensors (Fig 1c) are provided and utilized in similarity, entropy, and controllability calculations (see Numeric Representations of Hypergraphs in S1 File). Such tensor based calculations are not currently supported for non-uniform hypergraphs and will be pursued in the future. However, both uniform and non-uniform hypergraphs expand to pairwise structures (Fig 1d, see Hypergraph Expansions in S1 File). HAT contains hypergraphs expansions to generate clique expansions, star expansions, and line graphs. These representations facilitate indirect hypergraph similarity and entropy measures for non-uniform hypergraphs. Each hypergraph expansion has unique adjacency, degree, Laplacian, and normalized Laplacian matrices [27–29].

## Characteristic structural properties

The following structural properties of hypergraphs are computed: average distance between nodes is computed based on [13] (see Hypergraph Structural Properties in S1 File, Equation S1); the clustering coefficient is calculated based on [11] (Equation S2); hypergraph centrality is measured according to methods in [30, 31], which employ a variety of techniques to solve the nonlinear eigenvalue problem. For a uniform hypergraph, entropy is computed according to [11], which is defined based on the higher-order singular values of the Laplacian tensor (Equation S3). For non-uniform hypergraphs, standard graph entropy measures are applied to the aforementioned hypergraph expansions.

## Controllability

Hypergraph controllability refers to the ability to steer the underlying system of a hypergraph to a desired state by manipulating a subset of nodes (often referred to as driver nodes) [12]. For a uniform hypergraph, the minimum number of driver nodes required for controllability can be computed using the generalized Kalman's rank condition (see Hypergraph Controllability in S1 File, Equation S7). HAT is the first software to analyze controllability properties of hypergraphs.

## Similarity measures

Hypergraph similarity is measured according to the recent work [13], which distinguishes direct and indirect hypergraph similarity measures. Direct measures utilize tensor representations of uniform hypergraphs; indirect measures utilize graph similarity measures applied to hypergraph expansions. A series of structural and feature-based hypergraph similarity measures, including the Hamming Distance, the Jaccard Index, spectral measures, and centrality measures are provided (see Hypergraph Similarity Measures in S1 File, Equation S4 and S5). HAT is the first software to implement hypergraph similarity using a tensor representation based on the novel methods in [13].

## Results

Methods contained in HAT were utilized to examine Pore-C data (Fig 1a, see Pore-C in S1 File) [10]. Hypergraphs were constructed from Pore-C data from multiple cell types. Hypergraph similarity measures were employed to compare the structural similarity between different regions of the genome and cell types. In terms of biological implications, hypergraph entropy of chromosome structure has identified bifurcation points that determined cell fate over the course of a cell reprogramming experiment, which remained unidentified with a graph-theoretic approach [11]. Hypergraph analysis was also integrated with other sequencing modalities to identify transcriptional clusters and elucidate the higher-order organization of the genome [10].

In addition to examining Pore-C data, HAT was also used to quantify the activity of hypothalamic neurons monitored during a mouse feeding, fasting, and re-feeding experiment [32]. When constructing graph and hypergraph representations of the neuronal network within the hypothalamus for each phase of the experiment, hypergraph entropy proved to be a better indicator of changes in neuronal activity compared to graph entropy [11]. A similar result was also observed from a controllability/observability perspective under the same setting [12, 33].

Other applications of HAT include detecting influential hubs in social networks [34, 35], gaining insights into the stability and robustness of biochemical reaction networks [36, 37], identifying keystone species in ecological networks [38], and pinpointing control targets in epidemiological networks [39].

## Discussion

Hypergraphs can represent multi-way relationships unambiguously. The computational methods provided in HAT include hypergraph controllability and similarity measures from a tensor-based perspective. Additionally, the inclusion of tensor-based hypergraph structural properties (i.e., entropy and centrality), the association of multi-correlations with hypergraphs, and the integration of previously implemented graph expansion and visualization techniques within one software is an advancement over previously disjoint implementations. Therefore, HAT can advance the study of multi-way interactions in the genome or other complex biological systems.

## Supporting information

**S1 File. Supplementary information for HAT.**
(PDF)

## Acknowledgments

We would like to thank Dr. Frederick Leve at the Air Force Office of Scientific Research (AFOSR) for support and encouragement. We would also like to thank the two referees for their constructive comments, which led to a significant improvement of the article.

## Author Contributions

**Conceptualization:** Joshua Pickard, Indika Rajapakse.

**Formal analysis:** Joshua Pickard.

**Funding acquisition:** Indika Rajapakse.

**Investigation:** Can Chen, Indika Rajapakse.

**Methodology:** Joshua Pickard, Can Chen, Amit Surana, Anthony Bloch, Indika Rajapakse.

**Project administration:** Joshua Pickard, Indika Rajapakse.

**Resources:** Indika Rajapakse.

**Software:** Joshua Pickard, Can Chen, Rahmy Salman, Cooper Stansbury, Sion Kim, Amit Surana, Indika Rajapakse.

**Supervision:** Amit Surana, Anthony Bloch, Indika Rajapakse.

**Validation:** Joshua Pickard, Can Chen, Indika Rajapakse.

**Visualization:** Joshua Pickard, Can Chen, Rahmy Salman, Cooper Stansbury, Sion Kim, Indika Rajapakse.

**Writing – original draft:** Joshua Pickard, Can Chen, Cooper Stansbury, Sion Kim, Amit Surana, Anthony Bloch, Indika Rajapakse.

**Writing – review & editing:** Joshua Pickard, Can Chen, Sion Kim, Indika Rajapakse.

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
