## [Decision Letter · Decision Letter 0]

5 Apr 2023

Dear Dr Rajapakse,

Thank you very much for submitting your manuscript "HAT: Hypergraph Analysis Toolbox" for consideration at PLOS Computational Biology.

As with all papers reviewed by the journal, your manuscript was reviewed by members of the editorial board and by several independent reviewers. In light of the reviews (below this email), we would like to invite the resubmission of a significantly-revised version that takes into account the reviewers' comments.

Both reviewers have concerns about the paper. In particular, Reviewer #2 indicates that it is unclear which specific novel methods from the provided list the authors have developed.

We cannot make any decision about publication until we have seen the revised manuscript and your response to the reviewers' comments. Your revised manuscript is also likely to be sent to reviewers for further evaluation.

Sincerely,

Mark Alber, Ph.D.

Section Editor

PLOS Computational Biology

Mark Alber

Section Editor

PLOS Computational Biology

Reviewer's Responses to Questions

**Comments to the Authors:**

Reviewer #1: In this manuscript, the authors present a novel toolbox to work with hypergraphs, named Hypergraph Analysis Toolbox (HAT). I find the work novel and timely, and I think it can be of potential interest to many scientists working in biology and ecology from the perspective of complex science.

I found the references adequate, and the methodology well presented and explained. The authors have previously developed a code that is publicly available and documented, ready to be used. Given the interest of the subject (the study of many-way interactions in complex networks) and the few tools available, I consider this work worthy of publication in PlosComputational Biology.

I have however some minor corrections and some suggestions that I think can improve the manuscript before it is published.

L 28: split-pool tagmentation (SPRITE) -> Do the authors mean split-pool tag extension? As it is described in the publication where it is presented: SPRITE: a genome-wide method for mapping higher-order 3D interactions in the nucleus using combinatorial split-and-pool barcoding https://www.nature.com/articles/s41596-021-00633-y

L45 to 52: This paragraph could benefit from a little more work. I understand the authors are familiar with how different hypergraph tools were used in the case of the Pore-C data to unveil the higher-order organization of the genome, which is a nice example. However, they just mention other examples from social to ecological networks (references 20 to 27). I think a couple of sentences explaining how these tools were applied in these different systems and what was measured in them could improve the demonstration of the relevance of using hypergraphs in the study of complex systems.

L56 In reference 34 "Dengyong Zhou, Jiayuan Huang, and Bernhard Schölkopf. Beyond pairwise classification and clustering using hypergraphs. 2005." the name of the journal (or the book editorial etc) is missing.

L57 There is an error in the enumeration, the number 3 is repeated. It now reads "HAT targets the following hypergraph features and problems: 1[..], 2[..], 3[..], 3[..], and 4[..]" (it should be 1,2,3,4,5)

L60-L66: I encountered a problem, not in the text but running the code of HAT. For what it's worth I'll give a little detail: When trying to obtain the hypergraph of the Karate Club Network, the hyperedges2IM(hyperedges) function is not generating the incidence matrix properly. I fixed the threshold at 0.995 to only obtain 45 hyperedges and then generated the Incidence matrix. However, instead of including 1s and 0s in the proper places it fills the matrix with ones in the upper 35 rows or so, and 0s in the lower 10 rows. I'm using python 3.11, numpy1.23.5 , networkx 2.8.4, and matplotlib 3.7.1.

Characteristic Structural Properties and Similarity Measures: I highly recommend including the equations used to compute these metrics in the supplementary information of this manuscript (or even in a table in the main text). As it is now one needs to go to the original papers where they are presented, and in my case, the institution where I work is not subscribed to some of these journals. It is possible to read them in the arxiv version, but it can be confusing if the authors modified the equation numbering. Hence, in order to improve the clarity of this work, I think it is really important to include the equation of what is being measured together in the same manuscript where the toolkit is presented.

Reviewer #2: The software article “HAT: Hypergraph Analysis Toolbox” presents a tool for analysis of multi-way interactions and higher-order structures, addressing hypergraph construction (from experimental data) and respective analysis of structural and dynamic graph properties.

Although a list of hypergraph libraries is provided, the authors have not mentioned the “HyperG” package in R, that covers a broad variety of tools provided also by HAT. It would be very useful to maybe point to the features that HAT has additional to the already available algorithms. Morover, the author note that HAT “contains well-studied and novel mathematical methods for hypergraph analysis” , however from the provided list it is unclear to me which is the novel method that they have developed, or does this refer only to implementing a novel method – I assume the authors refer to controllability? In both cases, giving a mathematical description of the methods implemented would be useful for the users.

The authors have also chosen to demonstrate the method on a single experimental data set – if that is the case, then it would be useful to assess the advantage of the provided software over existing ones by demonstrating to the readers also with the respective analysis of the data set, and maybe suggesting possible biological implications that can result from analyzing the structural or dynamical properties of the constructed hypergraph.

**Have the authors made all data and (if applicable) computational code underlying the findings in their manuscript fully available?**

Reviewer #1: Yes

Reviewer #2: Yes

PLOS authors have the option to publish the peer review history of their article (what does this mean?). If published, this will include your full peer review and any attached files.

Reviewer #1: **Yes: **Virginia Domínguez-García

Reviewer #2: No
---

## [Decision Letter · Decision Letter 1]

16 May 2023

Dear Dr Rajapakse,

We are pleased to inform you that your manuscript 'HAT: Hypergraph Analysis Toolbox' has been provisionally accepted for publication in PLOS Computational Biology.

Best regards,

Mark Alber, Ph.D.

Section Editor

PLOS Computational Biology

Mark Alber

Section Editor

PLOS Computational Biology

Reviewer's Responses to Questions

**Comments to the Authors:**

Reviewer #1: I thank the authors for taking the time to implement the recommendations of both referees.

I believe the improved manuscript is ready for publication.

Sadly, I could not check if I still have problems with the corrected bug, due to time limitation, but I'll try to look into it in the GitHub page.

Reviewer #2: The authors have addressed the comments and suggestions raised to the previous version of the manuscript in great detail and I would like to thank them for the effort in this. I am happy to recommend the manuscript for publication.

**Have the authors made all data and (if applicable) computational code underlying the findings in their manuscript fully available?**

Reviewer #1: Yes

Reviewer #2: Yes

PLOS authors have the option to publish the peer review history of their article (what does this mean?). If published, this will include your full peer review and any attached files.

Reviewer #1: **Yes: **Virginia Domínguez-García

Reviewer #2: No

---

## [Editor Report · Acceptance letter]

1 Jun 2023

PCOMPBIOL-D-22-01881R1 

HAT: Hypergraph Analysis Toolbox

Dear Dr Rajapakse,

I am pleased to inform you that your manuscript has been formally accepted for publication in PLOS Computational Biology. Your manuscript is now with our production department and you will be notified of the publication date in due course.

With kind regards,

Anita Estes
